# The Role of NETosis and Complement Activation in COVID-19-Associated Coagulopathies

**DOI:** 10.3390/biomedicines11051371

**Published:** 2023-05-05

**Authors:** Emily Parissa Ghanbari, Kai Jakobs, Marianna Puccini, Leander Reinshagen, Julian Friebel, Arash Haghikia, Nicolle Kränkel, Ulf Landmesser, Ursula Rauch-Kröhnert

**Affiliations:** 1Department of Cardiology, Charité—Universitätsmedizin Berlin, Corporate Member of Freie Universität Berlin and Humboldt-Universität zu Berlin, 12203 Berlin, Germany; 2DZHK (German Centre for Cardiovascular Research), Partner Site Berlin, 10785 Berlin, Germany; 3Berlin Institute of Health at Charité—Universitätsmedizin Berlin, Charitéplatz 1, 10117 Berlin, Germany

**Keywords:** NETosis, coagulation, complement, COVID-19

## Abstract

Inflammation-induced coagulopathy is a common complication associated with coronavirus disease 2019 (COVID-19). We aim to evaluate the association of NETosis and complement markers with each other as well as their association with thrombogenicity and disease severity in COVID-19. The study included hospitalized patients with an acute respiratory infection: patients with SARS-CoV2 infection (COVpos, *n* = 47) or either pneumonia or infection-triggered acute exacerbated COPD (COVneg, *n* = 36). Our results show that NETosis, coagulation, and platelets, as well as complement markers, were significantly increased in COVpos patients, especially in severely ill COVpos patients. NETosis marker MPO/DNA complexes correlated with coagulation, platelet, and complement markers only in COVpos. Severely ill COVpos patients showed an association between complement C3 and SOFA (R = 0.48; *p* ≤ 0.028), C5 and SOFA (R = 0.46; *p* ≤ 0.038), and C5b-9 and SOFA (R = 0.44; *p* ≤ 0.046). This study provides further evidence that NETosis and the complement system are key players in COVID-19 inflammation and clinical severity. Unlike previous studies that found NETosis and complement markers to be elevated in COVID-19 patients compared to healthy controls, our findings show that this characteristic distinguishes COVID-19 from other pulmonary infectious diseases. Based on our results, we propose that COVID-19 patients at high risk for immunothrombosis could be identified via elevated complement markers such as C5.

## 1. Introduction

Coronavirus disease 2019 (COVID-19) has been associated with an increase in thromboembolic events [1,2,3]. Clinical research shows that patients show an elevated risk of deep vein thrombosis by 46%, pulmonary embolism by 24%, myocardial injury by 20%, and disseminated intravasal coagulopathy by 3% [3,4]. This state of hypercoagulability has been largely assigned to inflammation-induced coagulopathy (immunothrombosis), referring to the hyperactivation of the coagulation system due to the innate immune response upon infection [1,2,3,5]. In a systematic review and meta-analysis, COVID-19 patients with thromboembolism show a higher mortality rate of 23% in comparison to those without thromboembolism with 13% [6], which illustrates the importance of identifying patients at high risk for immunothrombosis.

Upon infection with severe acute respiratory syndrome coronavirus-2 (SARS-CoV2), the resulting local inflammation in endothelial cells leads to apoptosis and the release of inflammatory cytokines. Inflammation in the alveoli has been shown to result in pulmonary edema, systemic hyperinflammation, and intravascular coagulopathy. Previous research has associated the formation of neutrophil extracellular traps (NETs) with microvascular and macrovascular thrombosis in COVID-19, despite NETs having no previous association with viral infections [3,7].

As a part of the innate immune response, NETs aim to capture and kill pathogens, interacting with the complement and coagulation systems in doing so. NETs are actively released from neutrophils into the extracellular space during what is termed ‘NETosis’. They are comprised of decondensed neutrophilic DNA covered with histones, oxidant enzymes, and antimicrobial proteins such as myeloperoxidase (MPO) and neutrophil elastase (NE) [8,9,10]. Previous research has shown that NETs act as a platform for thrombogenesis by activating platelets, tissue factor (TF), and factor XII (FXII), and complement activation by activating complement-mediated cell lysis. Fibrin fibers strengthen NETs to capture pathogens, while complement-mediated cell opsonization and lysis reinforce NETs antimicrobial properties [11].

The coagulation system is divided into platelet aggregation (primary) and the coagulation cascade (secondary hemostasis), which are both stimulated by NETs. TF is the main initiator of blood coagulation [12,13]. TF-FVIIa complex activates factor 10 (FX), which cleaves prothrombin into thrombin. Thrombin is able to cleave fibrinogen into active fibrin. Fibrin networks around the platelets then build the thrombus [11,14,15].

Activated neutrophils stimulate the extrinsic coagulation pathway by upregulating TF mRNA and releasing TF on their NETs [9,11,16]. NETs also induce tissue factor pathway inhibitor (TFPI) degradation as the main extrinsic pathway inhibitor [7,17]. The intrinsic coagulation pathway is stimulated when negatively charged NETs directly bind to and activate the coagulation factor FXII, which is otherwise activated via negatively charged collagen fibers on the endothelial wall [18]. NETs also bind to the von Willebrand Factor (vWF), which provides a substrate for platelet adhesion. NET-platelet complexes act as scaffolds for thrombus formation [7,19].

Positive feedback loops between NET and the coagulation system predispose to a dysregulated thromboinflammatory response upon excessive inflammation during the cytokine storm. The resulting fibrin structure on NETs reinforces its structure that captures pathogens and increases fibrin resistance to plasmin-induced fibrinolysis, which increases thrombogenicity in COVID-19 [11,16,18].

The complement system refers to the cascade-like activation of complement proteins via serine proteases, which results in the formation of the membrane attack complex (MAC, C5b-C9 complex). Classified as a part of the innate immune system, C5b-9 creates a pore in the cell membrane through which metabolites and small proteins diffuse freely, which results in cell lysis. The generation of chemoattractants, C3a and C5a, leads to the recruitment and activation of neutrophils, as well as their adhesion to the lung epithelium [20]. By binding to the C5a-receptor on the neutrophil surface, C5a upregulates the expression of immune receptors (e.g., Toll-like receptors) and complement receptors [21]. MPO is also able to cleave C5 into C5a- and C5b-like active fragments. [11,22].

Platelet activation can be achieved by the insertion of the C5b-9 complex into the membrane [23], C1q binding to C1qR on the membrane [11], and platelet responsiveness to C3 [24]. Platelets activate the classical and alternative complement pathways on their membranes. Thrombin cleaves and activates C3 and C5 into its active components, which results in the anaphylatoxins C3a and C5a recruiting neutrophils. Active coagulation factors FXa, FXIa, and plasmin also generate C3a and C5a [11].

In this study, we aim to evaluate the role of NETosis and complement activation in COVID-19 disease severity, as the extent of their involvement is not yet fully understood. We evaluated NETosis-associated markers, i.e., myeloperoxidase/deoxyribonucleic (MPO/DNA) complexes and MPO, as well as complement proteins, i.e., C3, C5, C5a, and C5b-9 (MAC) and their association with each other, as well as their association with disease severity determined via SOFA score.

## 2. Materials and Methods

### 2.1. Study Design and Subjects

The study included 83 patients who were hospitalized with a respiratory tract infection, patients with COVID-19 (COVpos, *n* = 47), or patients with either pneumonia or infection-triggered acute exacerbated chronic obstructive pulmonary disease (COPD) (COVneg, *n* = 36) in Campus Benjamin Franklin, Charité Universitätsmedizin Berlin. COVID status was individually confirmed via polymerase chain reaction (PCR). Severely ill patients (*n* = 27) were characterized as having respiratory distress (≥30 breaths/min), oxygen saturation ≤ 93% at rest, and/or arterial partial pressure (PaO_2_) or fraction of inspired oxygen (FiO_2_) ≤ 300 mmHg according to the Diagnosis and Treatment Protocol for Novel Coronavirus Pneumonia [25]. All patients were aged 18 years or older. Patients with a known hematological or hemostatic disease, coagulopathy, or acute bleeding event, or those on dual antiplatelet therapy were not included in this study. Patients were recruited between May 2020 and May 2021. The study was approved by the local ethics committee (EA2/066/20, EA4/147/15) and conducted in compliance with the 1964 Declaration of Helsinki and its amendments and the Principles of Good Clinical Practice by the International Council for Harmonization 1996.

### 2.2. Data Collection

The baseline patient characteristics were taken from the hospital’s electronic medical records. The sepsis-related organ failure assessment (SOFA) score and the simplified acute physiology score II (SAPS II) score were assessed at the time of inclusion.

### 2.3. Blood Sampling

Blood was sampled within four days of patient hospitalization. Patient’s blood was drawn from the cubital veins using citrate tubes (Sarstedt-Monovette, Nümbrecht, Germany). Whole blood was separated for the experiments requiring plasma by centrifugation (1200× *g*, 10 min, room temperature) and stored at −80 °C for further analysis.

### 2.4. ELISA

The plasma concentrations of MPO, NETs (MPO/DNA complexes), tissue factor pathway inhibitor (TFPI), thrombin–antithrombin (TAT) complexes, tissue plasminogen activator (tPA), tissue factor (TF), vWF, beta-defensin 1, and complement components C3, C5, C5a, and sC5b-9 were measured via enzyme-linked immunosorbent assay (ELISA). Assays used were the High-Sensitivity Myeloperoxidase Human Assay Kit (Aviscera Bioscience, Santa Clara, CA, USA), Human Neutrophil extracellular traps ELISA Kit (Bioassay Technology Laboratory, Zheijang, China), Human TFPI Quantikine ELISA Kit (R&D Systems, Minneapolis, MN, USA), Human tPA ELISA Kit, Human TAT complex ELISA Kit, Tissue Factor ELISA Kit and Human VWF ELISA Kit (AssayMax, St. Charles, MO, USA), ELISA Kit for Defensin Beta 1, ELISA Kit for Complement Component 3, ELISA Kit for Complement Component 5, ELISA Kit for Complement Component 5a, and ELISA Kit for Terminal Complement Complex C5b-9 (Cloud-Clone Corp., Katy, TX, USA). ELISA was performed according to the manufacturer’s instructions and measured using the Tecan Infinite 200Pro (Tecan Group, Mannedorf, Switzerland). TF activity levels were determined using the Human Tissue Factor Chromogenic AssaySense Activity Assay Kit (AssayMax, St. Charles, MO, USA) and measured using the Tecan Infinite 200Pro (Tecan Group, Mannedorf, Switzerland).

### 2.5. Statistical Analysis

All tests were two-sided; 95% confidence intervals were used, and a *p* value of less than 0.05 was considered significant. All tests were two-sided and non-parametric unless stated otherwise. Continuous variables are presented as median values with interquartile ranges; categorical variables are presented as percentage values. Differences between the two groups were evaluated either with the Mann–Whitney U test or with the chi-squared test; correlations were made with Spearman’s test. Statistical analyses were performed using IBM SPSS Statistics 27 software, and graphs were generated with GraphPad Prism 8.

## 3. Results

### 3.1. Patient Characteristics

Out of the patients recruited in our study who were hospitalized due to respiratory illness, more COVpos patients were classified as severely ill and died during hospitalization. Patients in the COVpos cohort received higher anticoagulation according to the local standard of treatment based on the topical literature. Oral glucocorticoids and inhalative bronchodilators were administered more often in the COVpos group. More COVneg patients were diagnosed with chronic obstructive lung disease (COPD) or asthma bronchial (Table 1). No patients received a vaccination against SARS-CoV2 prior to study participation.

### 3.2. Increased NETosis, Coagulation, Platelet, and Complement Markers in COVpos

NETosis markers MPO and MPO/DNA complex concentrations were significantly higher in COVpos groups. Thus, patients with COVID-19, but not other respiratory infections, exhibited higher levels of NETosis markers (Figure 1).

COVpos showed significantly higher levels of tPA and TAT complexes (TAT) (Table 1). TF pathway proteins were also significantly elevated in COVpos, i.e., TF protein, TF activity, and tissue factor pathway inhibitor (TFPI) showed a significant elevation in COVpos (Table 1, Figure 2). The endothelial marker vWF was also increased in COVpos, whereas beta-defensin 1 did not differ significantly between groups (Table 1).

Complement component C5b-9 (membrane attack complex, MAC) was significantly increased in COVpos (Figure 3A). C3a, C5, and C5a did not show a significant difference between groups (Figure 3B–D).

### 3.3. Increased NETosis, Complement, and Coagulation Markers in Severely Ill COVpos

Severely ill COVpos patients had significantly increased levels of MPO/DNA complexes, C5a, C5b-9, and vWF (Figure 4). Deceased COVpos patients further displayed increased levels of C5 [133.3 (78.9; 158.5) mg/l vs. 61.6 (41.3; 103.4) mg/l; *p* ≤ 0.034], beta-defensin 1 [4.0 (1.1; 10) ng/mL vs. 1.1 (0.8; 1.7) ng/mL; *p* ≤ 0.023], and TFPI [172.0 (130.6; 256.0) ng/mL vs. 105.7 (67.1; 160.6) ng/mL; *p* ≤ 0.01].

### 3.4. NETosis Markers Associated with Complement in COVpos

MPO/DNA complexes correlated with complement C5 (R = 0.4; *p* ≤ 0.020) and C5b-9, respectively, in COVpos patients (R = 0.54; *p* ≤ 0.001) (Figure 5).

### 3.5. Complement Associated with Disease Severity in COVpos

Severely ill COVpos patients (*n* = 21) showed an association between C3 and SOFA, C5 and SOFA, and C5b-9 and SOFA (Figure 6), whereas COVneg did not. Severe COVpos patients also showed a correlation between C5 and SAPSII (R = 0.58; *p* ≤ 0.005) and C5a and SAPSII (R = 0.51; *p* ≤ 0.017), respectively.

### 3.6. Beta-Defensin 1 Associated with Complement and Disease Severity in COVpos

In severely ill COVpos patients, beta-defensin 1 showed a correlation with complement C3 (R = 0.48; *p* ≤ 0.027) and C5 (R = 0.82; *p* ≤ 0.001). Disease severity via SOFA (R = 0.53; *p* ≤ 0.013) and SAPSII (R = 0.72; *p* ≤ 0.001) also showed a positive correlation with beta-defensin 1 (Figure 7).

## 4. Discussion

### 4.1. Central Findings

The central findings of our study are as follows:NETosis and complement markers are higher in COVpos than in COVneg patients with acute respiratory disease.NETosis and complement markers were higher in severely ill COVpos patients.Increased complement activation markers were associated with a higher SOFA and SAPSII score.

Our findings suggest an association between increased markers for NETosis and complement activation with disease severity in patients with COVID-19, but not in patients with respiratory tract infections that are unrelated to COVID-19.

### 4.2. Higher NETosis Markers Are Associated with Complement Activation Only in COVID-19

We observed higher levels of NETosis markers, i.e., MPO and MPO/DNA complexes, in COVID-19 patients overall, which is in line with previous research [8,26,27]. NETosis has been widely associated with the pathophysiology behind severe COVID-19, leading to immunothrombotic events and respiratory failure [3,7]. We also showed that there was an association between NETosis and complement in COVID-19 patients but not in patients with non-COVID-19-related pulmonary infections, a distinction that has not yet been reported by previous research. Previous studies have also linked NETosis and complement to the pathophysiology behind COVID-19 disease severity [9,17].

We were not able to show a direct correlation between NETosis markers and the SOFA score in COVID-19 patients. Previous research has implicated NET production as a predictor of disease severity and clinical outcome mainly during severe COVID-19 [28]. This may provide an explanation for our findings, as the majority of patients recruited in this study were moderately ill.

### 4.3. Coagulation Markers Are Increased in COVID-19 Patients

We have shown an increase in coagulation markers in our study, which has also been observed in previous research. The elevation of markers of TAT and tPA has been attributed to prognostic values in relation to the occurrence of thrombosis and disease severity of COVID-19 [29,30,31]. The correlation between NETosis and coagulation markers, which we present in our findings, has also previously been reported in relation to the SARS-CoV2 infection [9].

Our results further point to an upregulation of the TF pathway in COVpos in contrast to COVneg pulmonary-infected patients. It has been established that TF plays a role in inflammation [32], and its upregulation has been postulated to be associated with thrombus formation in COVID-19 (28). NETs have been described to carry TF on their surfaces and thereby stimulate coagulation in COVID-19 in vitro [9,16]. Other studies have shown that TF exposure in NETs is dependent on the type of stimulation used to induce NETosis, with several in vitro failures to show NETs displaying TF [11,17]. A study by Skendros et al. showed an increase in TF mRNA expression in neutrophils treated with COVID-19-derived platelet-rich plasma (PRP) and increased TF/NE staining in neutrophils of COVID-19 patients [9]. The experiments conducted seem to be of more qualitative than quantitative value, as there is no evidence for the overall significance of the TF expression of NETs. In our study, we were not able to directly correlate the increase in TF pathway markers with NETosis markers or disease severity.

COVID-19 patients had higher plasma TFPI levels than pulmonary-infected non-COVID-19 patients. An increase in TFPI has previously been shown in moderately ill COVID-19 patients compared to healthy subjects [33]. Previous research on NETs in a model system, on the other hand, showed that NETs cleave and inactivate TFPI via NE during thrombotic events [7,34]. Our results do not show a COVID-19-associated depletion of TFPI or a significant correlation between TFPI and NETosis markers.

### 4.4. COVID-19-Related Increase in vWF

The increase in endothelial marker vWF, which we describe in our results, is consistent with previous research on this subject [35,36,37,38]. Platelets have been shown to be hyperactivated in moderately and severely ill COVID-19 patients and thus promote microthrombosis and overall disease progression by immune activation [5,39,40,41]. We were not able to show a direct association between platelet and NETosis markers in our results, despite the interaction between NETs and vWF being implicated as an important mechanism behind the platelet hyperactivation contribution to COVID-19-related thrombotic complications [10,38].

### 4.5. Higher Complement Markers Are Associated with Disease Severity Only in COVID-19

The complement protein C3 was also found to be elevated in COVpos patients in comparison to healthy controls in a previous study [42,43], as was sC5b-9 [9,44,45]. The levels of complement C5 and C5a were also shown to be elevated in comparison to healthy controls [22,42,43,45]. Our results do not, however, provide evidence that increased levels of C3, C5, and C5a characterize by COVID-19 in contrast to other pulmonary infections. Our results indicate that among COVpos patients, increased levels of C5a and C5b-9 may serve as markers for severely ill patients, and increased levels of C5 may be characteristic of deceased patients.

Other studies have suggested an association between complement C3, C5, and C5b-9 activation and disease severity, as seen in our results. Zhang et al. [42] also showed that C3 and C5 levels were associated with disease severity determined via clinical presentation. Cugno et al. [45] showed an association between C5b-9 and disease severity, as characterized by a high viral load measured via real-time polymerase chain reaction (RT-PCR). However, a statistical correlation between complements C3, C5, and C5b-9 in COVID-19 patients with established clinical scores of SOFA and SAPSII has not yet been described.

We propose that COVID-19 at high risk for immunothrombosis could be identified by complement C5. This is based on the elevation of C5 measured in deceased COVpos patients in comparison to COVneg and the association of C5 with the SOFA score and SAPSII score only in COVpos. C5a-receptor 1 (C5aR1) blockade via PMX53, a cell-permeable, orally available, non-competitive antagonist, was effectively used in a cell culture model to reduce NET expression [9]. Previous research on C3 inhibition via compstatin Cp40 showed the successful disruption of TF expression of neutrophils following COVID-19 serum-induced complement activation in vitro [9]. Inhibition of C5 with Eculizumab has been established in paroxysmal nocturnal hemoglobinuria and could be further investigated to manage COVID-19 disease severity [46].

### 4.6. Beta-Defensin 1 Might Play a Role in NET-Associated Platelet Recruitment

In *Staphylococcus aureus* infections, NETs have been described as bacterial scaffolds that allow *S. aureus* to stay in place and grow. NETs have been shown to retain their function despite *S. aureus* colonization. *S. aureus* promotes NETosis by activating platelets, i.e., by binding to vWF or via its alpha-toxin. The alpha-toxin causes the secretion of beta-defensin 1 from platelets, which induces NET formation [34,35]. As previously mentioned, we were not able to show a direct association between platelet and NETosis markers; however, beta-defensin 1 did markedly show an association with complement and disease severity parameters only in COVID-19 patients. This prompts further research into this mechanism of immunothrombosis in relation to COVID-19, which has not yet been touched upon by previous research.

### 4.7. Limitations

The limited number of patients and the heterogeneity of the COVneg group make the study prone to selection bias. The majority of the patients included in this study were not vaccinated, as the study recruitment process took place before vaccinations were widely available to the public. Limited research on the topic shows that vaccination with two or more doses significantly reduces the risk of pulmonary embolism [47]. The Omicron variant has been attributed with higher transmission rates and milder symptoms than previous SARS-CoV2 variants, but has not been associated with a significant decrease in thromboembolic events [48].

In relation to adenovirus vector-based vaccinations, however, studies have reported enhanced inflammation and platelet activation markers, as well as increased thrombin generation in infected patients [49]. Another study showed that an increase in NETosis markers was associated with the development and severity of vaccine-induced immune thrombotic thrombocytopenia (VITT) following adenovirus-based vaccination [50]. This suggests that NETs continue to play a role in disease complications after adenovirus-vector-based vaccination; however, more research is needed to fully elucidate the role of NETosis and complement activation following SARS-CoV-2 adenovirus vector-based and messenger ribonucleic acid (mRNA) vaccination.

COVID-19 has shown an increase in thromboembolic complications in affected patients not only during infection, but also in the convalescence period following the infection [51,52,53]. It has been suggested that NETs play a role in the residual low-grade inflammation and endothelial activation following COVID-19, termed ‘Post-COVID-19 Syndrome’, increasing the risk of thrombotic and other complications [51,54]. For that reason, in addition to the aforementioned, further research is needed to evaluate the significance of NETosis biomarkers post-COVID-19 infection, along with autoimmune markers, such as anti-cyclic citrullinated peptide (ANCA), rheumatoid factor (RA), and anti-NET antibodies (ANETA) [54].

## 5. Conclusions

Our current findings corroborate previous evidence for NETosis and complement-driven inflammation in COVID-19 and their relevance toward disease severity. Whereas previous studies have provided evidence that NETosis and complement markers are higher in COVID-19 patients than in healthy COVneg controls. This study shows that this characteristic distinguishes COVID-19 from other pulmonary infectious diseases. As COVID-19 patients showed a correlation between several complement components and SOFA, we suggest that COVID-19 patients who were at high risk for immunothrombosis could be identified via elevated complement markers such as C5.

## Figures and Tables

**Figure 1 biomedicines-11-01371-f001:**
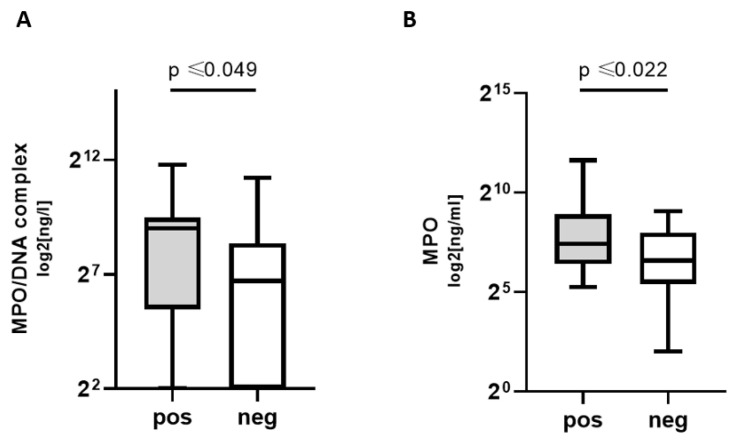
NETosis markers in COVpos patients. Levels of MPO (**A**) and MPO/DNA complexes (**B**) in COVpos (*n* = 47) and COVneg (*n* = 36) patients.

**Figure 2 biomedicines-11-01371-f002:**
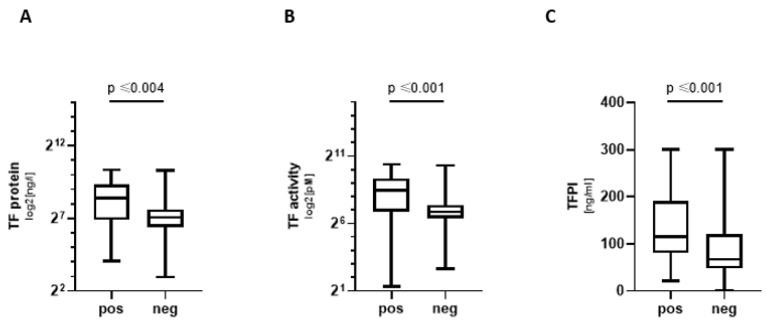
TF pathway markers in COVpos patients. Levels of TF concentration (**A**) and TF activity (**B**) and TFPI (**C**) in COVpos (*n* = 47) and COVneg (*n* = 36) patients.

**Figure 3 biomedicines-11-01371-f003:**
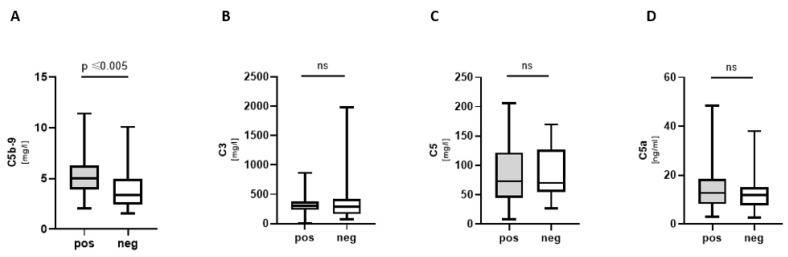
Complement components in COVpos patients. Levels of complement components (**A**–**D**) in COVpos (*n* = 47) and COVneg (*n* = 36) patients. Ns: non-significant.

**Figure 4 biomedicines-11-01371-f004:**
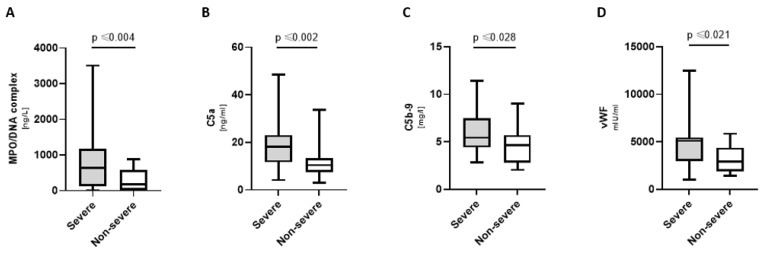
Markers elevated in severely ill COVpos patients. Levels of MPO/DNA complexes (**A**), complement components (**B**,**C**) and vWF (**D**) in severely ill (*n* = 21) and not severely ill (*n* = 36) COVpos patients.

**Figure 5 biomedicines-11-01371-f005:**
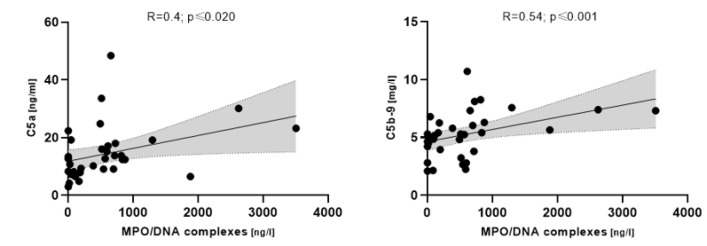
Correlation between MPO/DNA complexes and complement components in COVpos patients (*n* = 47).

**Figure 6 biomedicines-11-01371-f006:**
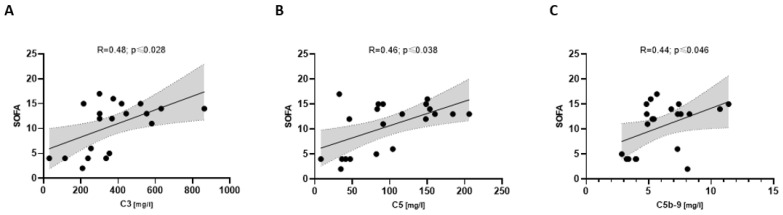
Correlation between complement components and the SOFA score (**A**–**C**) in COVpos patients (*n* = 47).

**Figure 7 biomedicines-11-01371-f007:**
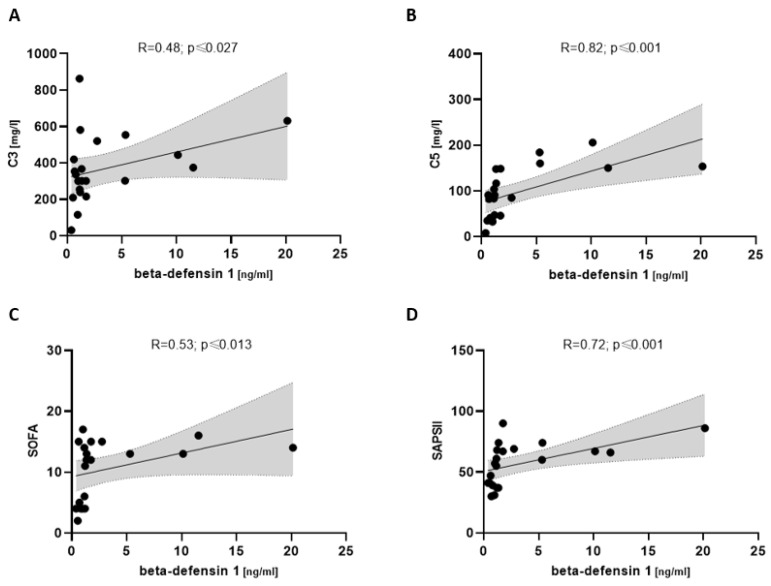
Correlation between beta-defensin 1 and complement components (**A**,**B**) and disease severity scores (**C**,**D**) in severely ill COVpos patients (*n* = 21).

**Table 1 biomedicines-11-01371-t001:** Baseline patient characteristics.

	COVpos *n* = 47	COVneg *n* = 36	Mann–Whitney U or Chi-Squared Test
**Demographics**			
Age (years) ^1^	70 [55; 78]	72.5 [58; 80.8]	0.380
Female (% per group)	31.9	41.7	0.362
Men (% per group)	68.1	58.3	0.362
BMI (kg/m^2) ^1^	26.5 [24.7; 30.4]	25 [23.1; 28.1]	0.140
Patients with severe illness	21	8	0.035
Deceased patients	7	0	0.010
**Pre-existing conditions**			
Coronary artery disease (% per group)	14.9	30.6	0.088
Peripheral artery disease (% per group)	4.3	30.6	0.001
Arterial hypertension (% per group)	63.8	66.7	0.789
Diabetes (% per group)	27.7	22.2	0.575
Dyslipidemia (% per group)	29.8	27.8	0.842
COPD or asthma bronchial (% per group)	6.4	36.1	0.001
**Concomitant medication**			
Acetylsalicylic acid (% per group)	38.3	30.6	0.466
Clopidogrel (% per group)	4.3	0	0.213
Prophylactic-dose anticoagulation (% per group)	25.5	58.3	0.003
Intermediate-dose anticoagulation (% per group)	27.7	8.3	0.028
Therapeutic-dose anticoagulation (% per group)	46.8	33.3	0.219
Statin (% per group)	25.5	25.0	0.956
ACE blocker (% per group)	25.5	36.1	0.301
Angiotensin II receptor blocker (% per group)	21.3	16.7	0.600
Beta blocker (% per group)	27.7	47.2	0.068
Aldosterone antagonist (% per group)	6.4	13.9	0.254
Diuretic (% per group)	38.3	47.2	0.417
Oral glucocorticoid (% per group)	55.3	19.4	0.001
Inhalative bronchodilator (% per group)	85.1	58.3	0.006
**Coagulation markers**			
tPA (ng/mL) ^1^	11.1 [7; 23.8]	7.3 [4.9; 10.6]	0.004
TAT (ng/mL) ^1^	4.9 [3.7; 7.8]	3.4 [2.5; 4.5]	0.0001
TF protein (ng/l) ^1^	343.3 [119.5; 647.7]	133.0 [84.2; 197.1]	0.004
TF activity (pM) ^1^	356.6 [117.3; 647.7]	117.6 [83.1; 167.3]	0.001
TFPI (ng/mL) ^1^	172.0 [130.6; 256.0]	105.7 [67.1; 160.6]	0.006
vWF (mlU/mL) ^1^	5100.8 [2953.3; 5472.6]	2903.9 [1897.2; 4393.7]	0.021
Beta-defensin 1 ^1^	1.2 [0.9; 1.7]	1.5 [0.8; 2.4]	0.693

^1^ median values with quartiles.

## Data Availability

Data from patients are not publicly available due to general data protection regulations. This could be available upon request.

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
