# Peer review of "The Role of NETosis and Complement Activation in COVID-19-Associated Coagulopathies"

_biomedicines, 2023, doi:10.3390/biomedicines11051371_

Round 1

Reviewer 1 Report

In this original paper, the authors compare several biomarkers of coagulation and complement activation in the plasma of patients suffering from COVID19 and other pneumonia.

Although well-witten, this manuscript has major limitations.

Major points:

1/The number of patients is limited.

2/COVIDneg group is not well defined and might be too heterogenous.

3/Patients with DAPT were excluded, why not patients under aspirin/clopidogrel monotherapy ? What about patients under anticoagulant therapy ? Why  33 to 47% of patients were under therapeutic dose anticoagulation ? Acute VTE ? Previous history of VTE ? Other ?(AF ?...) 

4/Type of plasma (citrate ?) and preparation methods (centrifugation steps) should be specified

5/Timing of blood sampling is not specified.

6/VWF and beta-defensin 1, although contained in platelet alpha granules are not specific platelet markers 

7/Pearson correlation is not appropriate for skewed values

Minor points:

1/COPD should be explained line 96.

2/line 96: infection-triggered... not infect-triggered

3/t-PA and TFPI methods of measurement are not specified

4/Table 1: numbers of patients receiving clopidogrel are missing

5/Major reference papers about COVID-associated patelet activation are missing (Manne BK, Blood 2020 & Zaid Y, Circ Res 2020 & Blood Adv 2021...)

Author Response

Response to Reviewer 1 Comments

Major points:

  1. The number of patients is limited.

This point is now mentioned as a limitation (line 328-329).

  1. COVIDneg group is not well defined and might be too heterogeneous.

This point is now mentioned as a limitation (line 328-329).

  1. Patients with DAPT were excluded, why not patients under aspirin/ clopidogrel monotherapy? What about patients under anticoagulant therapy? Why 33 to 47% of patients were under therapeutic dose anticoagulation? Acute VTE? Previous history of VTE? Other? (AF?...)

Patients under aspirin/ clopidogrel monotherapy were not excluded as we deemed the assessment of platelet function is not the main subject of this paper.

More COVpos patients received therapeutic-dose anticoagulation due to the local standard of treatment based on topical literature, which is now described in lines 155-156.

  1. Type of plasma (citrate ?) and preparation methods (centrifugation steps) should be specified.

The type of plasma and preparation methods are now specified in lines 120-123.

  1. Timing of blood sampling is not specified.

Timing of blood sampling is now specified in line 120.

  1. VWF and beta-defensin 1, although contained in platelet alpha granules are not specific platelet markers.

We agree with the reviewer. However, it was deemed negligible as platelet function is not the main topic of this paper. VWF was chosen due to its strong association as a COVID-19 marker (line 284-285). Beta-defensin 1 was chosen as its secretion has been associated with NETosis during S.aureus infection (line 319-320).

  1. Pearson correlation is not appropriate for skewed values.

Many thanks to the reviewer for drawing our attention towards this issue. The statistical methods used have been amended (line 149). We have re-calculated our correlations using Spearman’s method and have made according changes to the values in our results section and in the figures. Due to this, MPO/DNA complexes no longer show a significant correlation with tPA and vWF. Beta-defensin 1 additionally showed a correlation with complement and SOFA with Spearman’s test, which we have added in the results section.

Minor points:

  1. COPD should be explained line 96.

COPD is now described as chronic obstructive pulmonary disease in line 100.

  1. Line 96: infection-triggered … not infect-triggered

This has been corrected (line 99).

  1. tPA and TFPI methods of measurement are not specified.

tPA and TFPI methods of measurement have now been specified in lines 132-133.

  1. Table 1: numbers of patients receiving clopidogrel missing.

This information has been added to Table 1.

Reviewer 2 Report

Dear authors

Please add values in the Abstract.

As there are previous similar studies, please highlight novility of this study.

Some English and writing corrections are needed.

The number of samples is low.

Please check “study partipation”

In the discussion section, Coagulation markers are increased in COVID-19 patients should be compared with previous findings.

Please write microorganisms names (S. aureus) in italic.

The study has assessed important topic, however it will be more interesting for readers about treatment options of the disease or any preventive approach.

Please also provide statistics of the disease and risk factors.

Kind regards

Some English and writing corrections are needed.

Kind regards

Author Response

Response to Reviewer 2 Comments

  1. Please add values in the Abstract.

The correlation values between complement and SOFA have been added into the Abstract (line 20-21).

  1. As there are previous similar studies, please highlight the novelty of this study.

The novelty of the study lies in the choice of respiratory tract infected patients as the control group, rather than healthy subjects. Therefore, our findings suggest an association between increased markers for NETosis and complement activation with disease severity in patients with COVID-19, but not in patients with respiratory tract infection unrelated to COVID-19, which has not yet been shown by previous research. This is mentioned in the Abstract (line 22-25) and in the discussion (line 232-234).

  1. Some English and writing corrections are needed.

We have repeatedly checked for and corrected language and spelling mistakes.

  1. The number of samples is low.

This point is now mentioned as a limitation (line 328-329).

  1. Please check “study participation”.

Informed consent was obtained from all study participants, clarified in lines 373-374.

  1. In the discussion section, coagulation markers increased in COVID-19 should be compared with previous findings.

The coagulation markers measured in this study are tPA, TAT, TF protein, TF activity and TFPI (Table 1). The elevation of tPA and TAT related to COVID-19 is now discussed in lines 244-246. The increase of TF pathway proteins is discussed in lines 250-262.

  1. Please write microorganisms names (S. aureus) in italic.

This has been amended (line 316-318).

  1. The study has assessed important topic, however it will be more interesting for readers about treatment options of the disease or any preventive approach.

Based on our findings, we suggest that COVID-19 patients at high risk for immunothrombosis could be identified via elevated complement markers such as C5. We also mention in lines 311-313 that inhibition of C5 with Eculizumab has been established in paroxysmal nocturnal hemoglobinuria and could be further investigated to manage COVID-19 disease severity.

  1. Please provide statistics of the disease and risk factors.

Statistics of the disease and risk factors have been added. The incidence of several risk factors in COVID-19: DVT, PE, MI and DIC in lines 31-33. The mortality rate of COVID-19 patients with TE vs without TE has been added in lines 36-39.

Reviewer 3 Report

Thank you for the opportunity to review this interesting article. In general, the paper is well-structured and easy to follow. The authors aim to aim to evaluate the role of NETosis and complement activation in COVID-19 disease severity. They concluded that elevated complement markers such as C5 is associated with high risk for immunothrombosis in COVID-19 patients. This is a valuable submission that I recommend for publication with a few minor changes.

Minor concerns:

1. Line 16: Please provide the results of coagulation and platelet in Table 1.

2. Line 92: Study design and subjects: please add the hospital name(s) included in the study.

3. Line 109: Please indicate the anticoagulant (heparin? sodium citrate? K2EDTA?)

4.Line 113: Enzyme-linked immunosorbent assay (ELISA)

5.Line 2.4: Statistical analysis

6.Line 146: Please confirm the number of COVneg (37? or 36?). Line 96 showed that the number of COVneg is 37. The term COVIDneg and COVneg is not inconsistent.

7.Table 1. Please add the percentage of female

8.Line 284, line 285: Bacteria genus and species should be write in italic

9.In the text, reference numbers should be placed in square brackets [ ].

Minor editing of English language required

Author Response

Response to Reviewer 3 Comments

Minor concerns:

  1. Line 16: Please provide the results of coagulation and platelet in Table 1.

The results of coagulation and platelet are now provided in Table 1.

  1. Line 92: Study design and subjects: please add the hospital name included in the study.

The hospital name has now been included in lines 100-101.

  1. Line 109: Please indicate the anticoagulant.

The anticoagulant has now been added (line 121).

  1. Line 113: Enzyme-linked immunosorbent assay (ELISA)

Word has been capitalized (line 129).

  1. Line 2.4: Statistical analysis

Has been corrected (line 143).

  1. Line 146: Please confirm the number of COVneg (37? or 36?) Line 96 showed that the number of COVneg is 37. The term COVIDneg and COVneg is not consistent.

The number of COVneg is 36 and has been corrected in line 100. COVneg is now used consistently as an abbreviated term.

  1. Table 1. Please add the percentage of female

The percentage of female study participants has been added in Table 1.

  1. Line 284, line 285: Bacteria genus and species should be written in italics.

This has been amended (line 316-318).

  1. In the text, reference numbers should be placed in square brackets [].

This has been corrected.

Round 2

Reviewer 1 Report

The authors correctly addressed all the issues.

However they should pay attention to the way they refer to some markers, for example Line 289 and 316: tPA and vwf are both endothelial markers not coagulation or platelet markers, even if vwf is contained in alpha granules, plasma vwf is mainly derived from EC...

Even if the main topic of the manuscript is not platelet activation, the first 2 main papers on the subject should be added to ref 5 and 39 (PMID 32938299 and 32573711)

Author Response

Response to Reviewer 1 Comments

  1. Line 289 and 316: tPA and vwf are both endothelial markers not coagulation or platelet markers, even if vwf is contained in alpha granules, plasma vwf is mainly derived from EC....

vWF and tPA are now referred to ad endothelial markers throughout the manuscript.

  1. Even if the main topic of the manuscript is not platelet activation, the first 2 main papers on the subject should be added to ref 5 and 39 (PMID 32938299 and 32573711)

Papers by Manne et al. Blood 2020 PMID 32938299 and Zaid et al. Circ Res 2020 PMID 32573711 are now referenced in line 285.

Reviewer 3 Report

The authors have addressed all my questions and concerns.

Author Response

Thank you for your feedback!